# Creating Innovation in Achieving Sustainability: Halal-Friendly Sustainable Port

**Harlina Suzana Jaafar** [1,*] **, Mona Leza Abd Aziz** [2] **, Muhammad Razif Ahmad** [3] **and Nasruddin Faisol** [4]

1   Faculty of Business and Managemen, Universiti Teknologi MARA, Shah Alam 40450, Selangor, Malaysia
2   Johor Port Authority, Pasir Gudang 81700, Johor, Malaysia; monaleza@lpj.gov.my
3   Maharani Energy Gateway Sdn Bhd, Johor Bahru 81200, Johor, Malaysia; razifahmad@gmail.com
4   Faculty of Architecture, Planning and Surveying, Universiti Teknologi MARA,
    Shah Alam 40450, Selangor, Malaysia; nasru793@uitm.edu.my
*   Correspondence: harlinas@uitm.edu.my

**Abstract:** The expansion of liberalized trade has forced companies to consider the global market demand to stay competitive. Hence, ports have started to embrace sustainability practices in their activities throughout port operations. Various research has suggested that there is more innovation when sustainability is adopted as an integral part of their business activities. This study established a halal-friendly sustainable port concept and its implementation in meeting the objectives of sustainability practices to boost innovation. To embed sustainability within port organizations, it is vital to create an organizational culture that supports innovation and integrative thinking. Based on the qualitative data obtained from 38 port stakeholders in the southern of Malaysia, the respondents supported a halal-friendly sustainable port as a potential innovation in meeting the objectives of the sustainable practices. Four components that were found crucial for the proposed framework reflect the novelty of the research and its successful implementation.

**Keywords:** sustainability; innovation; ports; halal; environmental; supply chain

## 1. Introduction

The rise in green and sustainability brings a wave of change to the port and logistics industry [1–3]. Koilo [4] highlighted that practitioners are increasingly interested in sustainability issues in the maritime logistics and supply chain industry. This means that the international ports need to fulfill the requirements to be a sustainable port while being able to oblige to customer demand to stay competitive. In terms of economy, the increasing development of liberalization of trade and services has also called for companies to continuously be responsive to the global market demand to remain relevant. Hence, ports have started to embrace sustainability practices in their activities throughout port operations by placing priorities toward achieving economic, eco-social, and efficient operational goals [5–8]. Various research convinced that innovation is stimulated when sustainability is adopted as a fundamental part of their commercial activities [9–12].

In another stream of studies, several researchers emphasized that innovation could increase profits, help to cut costs [13–16], or enhance the quality of current process [17] and increase competitiveness [18]. Flint et al. [19] pointed out that innovation is significant to the logistics industry as it drives companies to be more competitive. However, the concept of innovation has been mainly disregarded in logistics research. They noted that many studies exploring innovation have put much emphasis on product innovation, especially innovation on high technology.

In another scenario, the concept of halal has increasingly attracted business widely. It is a concept gained from the Islamic principles that Muslims must consume halal products. Halal is obtained when all process, including the production and delivery of the products to the final consumers, comply with the Islamic principles. The requirements to be halal

are to ensure that the products are free from contamination with haram (forbidden by Islamic law) and hazardous products. As a result, the non-Muslims are becoming attracted to the safety and quality aspect of the halal products. The religious requirement has created the demand for halal products. The failure to consume halal products would result in a sin committed by the Muslims. From the business perspective, halal practice has been considered as a business strategy that would bring market expansion [20] as it addresses the growing number of Muslims and non-Muslims worldwide. The existing literature indicates the focus on producing the halal products from the manufacturing perspective. However, it is insufficient to reach the status of halal products when it reaches the final consumers [21–24]. Throughout these supply chain activities, several conditions could expose the halal products to the risk of contamination with haram and hazardous substances that would affect the halal status of the product when it reaches the final consumers. Those activities involve handling, storage, warehousing, and transportation, as well as sourcing and manufacturing.

Having considered the gaps and recent trends in logistics research and businesses, this study establishes the concept and framework of Halal-Friendly Sustainable Port (HFSP) due to its role as an international gateway that would complement the entire supply chain in a global context and identify how innovation is created in achieving sustainability in the port industry. Accordingly, this paper reviews the application of sustainability in the logistics and supply chain context by narrowing down to application of sustainability in the ports industry. The second part of the reviews explain the concept of halal supply chain and how innovation emerges from sustainability and halal supply chain implementation. The methodology was briefly explained before the findings were deliberated discussed.

## 2. Background of the Study

### 2.1. Sustainability in the Logistics and Supply Chain Industry

Sustainable supply chain management (SSCM) has become popular among the academics and industry experts in recent years [25,26]. It is reflected in the increasing number of publications debating the sustainability problems of supply chain networks in the last few years. It is generally agreed that sustainability comprises of an integration of three components that has always been referred to as social, environmental, and economic responsibilities in business disciplines [27] (see Figure 1). Elkington (1997) [28] popularized the three dimensions as the triple bottom line (TBL) principles (known as the three pillars: profit, planet, and people). It has also increasingly appeared as the key topic in supply chain management, especially in discussing its function in reducing the cost of operation, reducing waste, using more from less, and efficiency-oriented planning. Many organizations are beginning to rapidly adopt sustainable business practices to lower the cost of business operations so that the resources for the future will last longer and concurrently meet the calls of the governments, non-governmental organizations, and public pressure to promote the efficient use of resources. The optimization of operations has moved from a specific element of supply chain to the entire supply chain and making products of the greatest value and the lowest possible cost during the last two decades [29]. Several authors suggested that company achievement may be enhanced when sustainability is adopted along the supply chain systems [25,30–32].

Particularly in logistics, several researchers claimed that sustainability practices are not restricted to a single firm because the benefits or detrimental impacts can affect the entire product life cycle [33]. Dey et al. [34] highlighted that there are numerous areas where sustainability can be executed throughout the firm's logistics operations. Several authors agreed that logistical functions are critical in ensuring the execution of a sustainability strategy in the supply chain operations [35,36]. This is because logistical operations play a crucial role in managing the entire product movements through the supply chain, hence forming the impetus for the achievement of sustainability practice. Firms become accountable to their supply chains and, thus, influence them to inevitably quantify, control,

and reveal their own sustainability performance in addition to the whole supply chain sustainability achievement [37]. This includes various ways to reduce cost, improve time management, shorten activity processes, and improve the work quality of their employees.

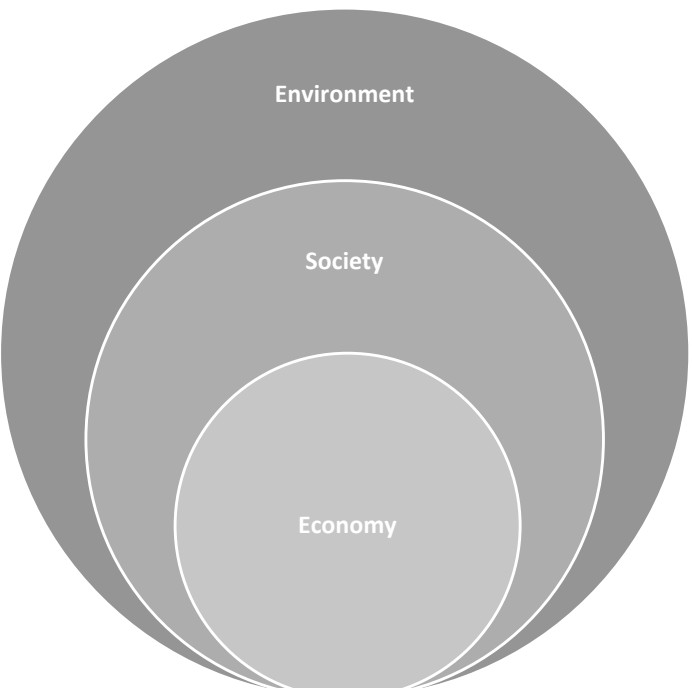

**Figure 1.** Nested Dependencies Model of Sustainability. Source: Doppelt, B. The Power of Sustainable Thinking: How to Create a Positive Future for the Climate, the Planet, Your Organization and Your Life; Routledge: New York, NY, USA, 2010; doi:10.4324/9781849773232.

Following these developments, the establishment of performance measurement on sustainability has grown as it promotes innovation, improves work processes, and increases efficiency while protecting the environment. Cost has been the most popular assessment criterion followed by customer, internal processes, innovativeness, flexibility, reliability, time, responsiveness, quality, asset management, efficiency, resource, output, and information [38].

Environmental dimension. For the past 20 years, the environmental aspect of supply chain management has received special attention. The environment seemed to be the focal point of the triple bottom line and has been highlighted due to global climate change and soaring energy prices. Since then, various terminologies have been used in several approaches of literature including green management, green supply chain management, green supply chain, sustainability, and sustainable supply chain management. Dubey et al. [25], who developed the World Class Sustainable Supply Chain Management (WCSSCM), highlighted that many authors consider the environment component to be comprised of elements, such as life cycle concept implementation, green product design, green packaging, green distribution, and warehousing, as well as conservation of natural resources. This is due to the need of eco-friendly practices, technologies, products, energy-efficient systems, and conversation methods. The shift from packaging to energy-efficient logistics in reducing global carbon footprints was also reflected as an environmental dimension of sustainability. Ji et al. [39] recommended various ways to reduce carbon footprints, such as improving the accuracy of demand forecasting, investing in carbon reduction technology, implementing smaller packaging, combining distribution, connecting with third-party logistics providers, adopting cross-docking networks, improving energy efficiency, shortening consumption time, combining design for ecology, and comprehensive reverse networks. To a certain extent, there has been an interchangeable usage of the

sustainability and environment terms by the researchers and industry practitioners. This misunderstanding was particularly common, especially when the terms were beginning to be adopted and theorized. However, researchers and practitioners have finally come to a consensus in terms of understanding and implementation of the sustainability term as the triple bottom line [40].

In the port industry, the port stakeholders, including the port authorities, policymakers, port users, and local communities, are becoming apprehensive of the environmental sustainability. They realized competitiveness and efficiency could be accelerated through the adoption of technical and process innovations that can address the key environmental issues and sustain quality standards, as well as ultimately improve efficiency and competitiveness [3]. However, in an environmental and economic perspective, Ashby et al. [41] and Halldorsson et al. [42] argued that the word "sustainability" has been exaggerated because it has included the unnecessary dimension of sustainability in supply chains.

Social dimension. In terms of the social values and ethics dimension, Dubey et al. [25] and Simoes et al. [43] claimed that the social component that consists of the relevant parties should be integrated during the planning process. Thus, the crucial features and relevant procedures could be incorporated during the design, plan, and operation stages of the social sustainable supply chains. Keating et al. [44] and Tencati et al. [45] underlined that the social component of SSCM is a part of corporate social responsibility (CSR). However, Seuring and Muller [7] and Dubey et al. [25] discovered that the social values and ethical components in SSCM is rather scarce and insufficiently addressed in the current literature. It lacks the behavioral component, especially in proving how the adoption of sustainability may affect management decision-making and, resultantly, innovation outcomes. They argued that most studies are based on empirical or mathematical models, multicriteria decision-making tools, as well as a lack of ethical dimension. The qualitative case studies, ethnography, and action research methods have received less attention among researchers when concentration has been given to more positivist approaches. Consequently, the influences of sustainable practices on social dimensions have been neglected.

Economic dimension. Various researchers consider profitability, strategic collaboration, information sharing, and logistics optimization as four indicators that are critical to accomplish economic stability. At the beginning stage of sustainability practices, the value enhancement of the entire system was considered as the most relevant to all stakeholders [46]. As a field of study, the economic dimension has also been considered as a failure to address issues related to sustainability [47]. Issues were examined in isolation with other facets of social responsibility [48]. Researchers, such as Min and Galle [49,50], explained that firms will only be sustainable when all three components (economic, social, and environmental are implemented. This is because the corporate financial benefits of the social and environmental components of SSCM would be realized in longer terms. In fact, previous studies found that supply chain collaboration and internal environmental activities are crucial in addressing all three dimensions of sustainable development: social, economic, and environmental [44,51–57].

### 2.2. Sustainability in the Port Industry

Ports play a role as crucial connectives within international logistics. Thus, they may become a chokepoint if they are inefficient to support the smooth flow process across borders. This is due to the complex role of a port, especially when it involves the movement of cargoes from land onto the ships or vice versa [58]. The ever-changing nature and role of port ownerships has led to the establishment of ports as service centers that coordinate the transport network and introduce many value-added services, particularly in the shipping operations (such as consolidation services, cross-docking operations, and one-stop center for meetings among the stakeholders) [59–61]. These new and vibrant roles of ports that drive port efficiency may attract more shippers that would, in turn, benefit the port authorities, service providers, customers, and other port stakeholders that may lead to the multiplier effect on the regional economy.

To achieve sustainability, ports play a crucial role in developing collaborations among the members of the port community, including warehouse operators, freight forwarders, and shipping agents, and ensuring excellent operational and logistics services [58]. The increasing competition among ports has forced ports to focus on fulfilling customer demand through the provision of numerous value-added services to expand its customer base. Such services may involve cargo security, types of consolidation, and cargo handling, among others [62,63]. The competitive activities in ports have spilled over to the whole of the supply chain activities, as the entities and services were connected to each other [64,65]. Accordingly, ports form the core components within the whole logistical chain in national freight routes [63,66].

In terms of services, the role of port cities as the focal points for imports and exports as well as the flow of goods across borders indicate the vital influence of ports to sustainable development [67–69]. In fact, the sustainability of port operations is highly reliant on the service quality and their customer satisfaction [70]. Yeo et al. [70] suggested that "failure or unreliability of port services can significantly influence port customers." Various research related to sustainable development of ports [5,7,71–73] provide a broad spectrum in the development of port strategies, practice implementation, policies, procedures, and activities to manage sustainability issues. It is even more complex to address the sustainability issues specific to economic, social, and environmental performance measures. Sloan [74] highlighted that measuring sustainability performance of supply chains is intrinsically complicated and multidimensional. A comprehensive supply chain performance measurement tool is deemed needed to evaluate and enhance the sustainability performance. To be sustainable, firms involved in risks [75] and the transformation toward sustainability entails innovative actions that bring about "creative destruction of unsustainable practices and their replacement with sustainable technologies and business models" leading to invention [46,76]. Even though it is quite risky in terms of the firm's survival, Venus [77] and Kim and Chiang [78] highlighted that the move toward sustainable ports may enhance port competitiveness. Therefore, firms may face negative consequences, such as a lack of attractiveness and increasing operational issues, if they fail to satisfy their customers [67,68].

In terms of CSR in seaports, Malik [79] highlighted that those studies on CSR in seaports are lacking compared to CSR studies in other contexts of economic sectors and in general business practice. The environmental aspect of seaport studies demonstrates a more general management or measures [80]. The issues covered under the environmental aspects include port pollution [81] and clean hinterland transport [82]. Economic issues have been dealt in the scope of port efficiency [83–86] or port hinterland optimization [87]. However, there has been a lack of focus on the social aspect of CSR research in seaport studies [88].

With the exception of Haralambides and Gujar [89] and Lam and Gu [90], studies addressing the sustainability agenda in seaports and its implementations are lacking. Haralambides and Gujar [89] and Lam and Gu [90] integrated both the economic and environmental objectives in determining sustainability practices.

### 2.3. Halal-Friendly Sustainable Port (HFSP) as an Innovation

The concept of halal supply chain. Halal is an Arabic term, Al-Halal (The Lawful), which means "permitted, with respect to which no restriction exists, and the doing of which the Law-Giver, Allah SWT has allowed." On the contrary, Haram or Al-Haram (The Prohibited) suggests that the "Law-Giver has absolutely prohibited. Anyone who commits is liable to incur the punishment in the Hereafter and a legal punishment in this world." Between halal and haram is Al-Makruh (The Detestable), which denotes "the disapproved by the Law-Giver but not very strongly." In the halal industry, the handling of food along with the logistics and supply chain process is deemed vital [91] as halal food production will be pointless if cleanliness of halal food is not maintained throughout the delivery process from the source of supply to the end consumers. A product can achieve halal status when all possible contamination caused by haram and hazardous products could be

avoided throughout its supply chain process [21–24] until, finally, it reaches the consumers. Thus, a halal process could be viewed from a supply chain perspective as a halal product could only be obtained when the entire activities throughout the supply chain process are dedicated only for halal products or separated from the haram and hazardous products [92]. This includes the source of supply, warehousing and storage, handling, manufacturing, and transportation activities.

The halal industry. With the significant increase in the Muslim population, the halal market is expected to reach USD 3.2 trillion by 2024 with a projected compound annual growth rate (CAGR) of 6.2% between 2018 to 2024 [93]. The State of Global Islamic Economy Report 2019/2020 declared that the Islamic economic sectors consist of halal food, modest fashion, media and creation, Muslim-friendly travel, halal pharmaceuticals, halal cosmetics, and Islamic finance. However, concerns about halal logistics practices are on products that might be exposed to contamination throughout the journey, such as food and beverages, pharmaceuticals, and cosmetics. Therefore, logistical activities, such as handling, transportation, storage, and retail, are deemed crucial. The most critical products are the cold chain products, involving food, beverages, and pharmaceuticals that could be contaminated if the temperature is not well controlled throughout the journey in the entire supply chain. The consolidation of halal and haram products may also lead to contamination that might affect the Muslim requirements. Since logistics is a derived demand, the emergence of halal logistics will result from the increasing demand of halal products.

Issues of contamination. It is common that food manufacturers always deal with food contamination during transportation and delivery to the retailers [94]. Thus, logistical companies must deal with the complexity of the delivery process of the fresh and finished food products. The delivery process may involve storage at the farm and warehouse, as well as loading and unloading onto the refrigerated trucks, containers, and a feeder ship, until the trucks reach the final destinations. The delivery process may be exposed to shipping delays and contamination due to fungi and bacteria, damages, spills, and breakages, and bad conditions during shipping, which may later cause contamination and food poisoning or food-borne illness. As food products are normally temperature-sensitive, possible contamination may occur throughout the delivery process, including packaging, processing, shipping, as well as storage and distribution. A lack of concern about the contamination may result in health hazards that can lead to varied symptoms, such as neurological, renal, and hepatic syndromes. Accordingly, logistics companies should adhere to strict safety and hygiene procedures and regulations as well as ensuring the employment of well-experienced and trained personnel in food safety. In fact, contamination, damages, breakages, and spills may also lead to substantial losses to the food manufacturers. Thus, special procedures need to be imposed throughout the entire delivery process.

Studies in halal supply chain. Rasi et al. [95] underlined that a halal supply chain comprises of four main components: halal procurement, halal manufacturing, halal distribution, and halal logistics. Halal procurement deals with inputs, by-products, and resources that halal because they are vital to ensure halal integrity. Some researchers noted that supplier selection is crucial in the absence of traceability tools due to complexity of halal supply chain [95,96]. They stressed that halal manufacturing should source for halal materials and the production process should adhere to halal procedures to produce halal outputs. Halal packaging, halal containers and halal logistics, on the other hand dealt with process activities, namely the process of organizing and protecting products and materials before the halal products are delivered to the customers [23].

In a review study on the definition of halal supply chain management, Khan et al. [97] emphasized that the current definitions of halal supply chain tend to include all aspects of halal of the entire supply chain and the conventional supply chain management. However, they claimed that the halal supply chain can be differentiated from the conventional supply chain in terms of flow of material, information, and capital in such a way that Halal and Toyyib (cleanliness) is extended to the consumption point. They also suggested that halal

should be based on a process-oriented approach and should be coordinated and monitored closely by the entities along the supply chain in the form of collaboration so that value and enhancement could be created and obtained. However, they viewed that researchers may categorize and define the halal supply chain according to their context.

Innovation in the logistics and supply chain industry. Innovation is described as the implementation of new combinations of product, process, and organizational innovations that would give new access to markets of suppliers or consumers [98]. It implies the way information, inspiration, and creativity is implemented to enhance value creation that encompasses the entire processes toward development of new and useful products. Innovation represents an idea that could be replicable and cost-effective in order to fulfil certain needs. In line with Schumpeter [99], Flint et al. [19] pointed out that innovation is not limited to technological or product inventions, but the concept of innovation could occur within services, processes, or any social system.

The economists have realized the significance of innovation impact on company performance and economic growth for decades [100]. Innovativeness has consistently become a significant determinant to higher firm performance [13,101–103] as it enables firms to offer a wide variety of valuable, rare, inimitable, and differentiated products [104]. Any logistics-related service that is viewed as novel and may assist customers can be considered as logistics innovation [19]. This may include any activities that may enhance operational efficiency as well as fulfill customer demand. From a supply chain management perspective, a firm is considered innovative when it has frequent invention of internal logistics-related processes and is able to generate new creative ideas in operational procedures. Panayides and Lun [105] emphasized that even the orientation of a firm's culture toward innovation is likely to improve supply chain performance. Bell et al. [106] highlighted that the key to innovation is to improve operational efficiency and enhance service effectiveness. This could be done through an integrated development of information and technology with a new marketing process and logistics. Bowersox et al. [107] noted that the attention given to cost savings and service improvements by the industry has reflected a significant need to consider innovation in the entire supply chain so that efficiency could be maximized and competitive pressure could be reduced. However, most past studies on certain aspects of innovation demonstrate a lack of attention to the supply chain innovation, particularly [108,109] service innovations.

Teece [110] argued that an organization can achieve long-term organizational performance when it has the capability to innovate. Several authors explained that innovation is created when the goods and services are considered as added value by the clients. Successful organizations that are flexible in terms of revitalizing its procedures and practices, as well as reviewing and creating new business process, are normally organizations that put innovation as top priority [111,112]. On the contrary, organizations that are rigid and inflexible may eventually lead to organizational failure [113]. Considering its importance, there is an urgent need for an organization to consider innovation capacity as one of the core competencies of their personnel. Calic et al. [114] described the development factor that may lead to long-term successful organizational sustainability. They claimed that sustainability constraints could be the impetus of the generation of new ideas and managerial quality and speed up the process.

Certification of halal supply chain. Halal supply chain standards have been widely accepted as international standard under the Standards and Metrology Institute for the Islamic Countries (SMIIC) through the completion of its development and published on the 9th of October 2020. These standards that were adopted from the Malaysian Standards (MS2400) covers transportation, warehousing, and retailing [115–117]. Having considered the global complexity of the production, the importance of international trade of halal products is deemed crucial. Therefore, this study aims to introduce the concept of HFSP and its implementation in meeting the objectives of sustainability practice, particularly the social objectives. This is because a balance of environmental, economic, and social responsibility initiatives is crucial in meeting the sustainable development goals (SDGs)

which should be adopted as a fundamental business practice. To embed sustainability within port organizations, there is a need to focus on creating an organizational culture that supports innovation and integrative thinking.

In Malaysia, with the exception of Aziz and Zailani [118], studies on halal-friendly ports are particularly lacking in the current literature [25]. The study by Aziz and Zailani [118] was also not well discussed, especially on the application of halal elements in the port sectors.

## 3. Research Methodology

Based on an ongoing study, this paper aims to develop a halal supply chain framework in the context of international trade that focuses on port as an international gateway. By referring to ports at the southern region of Malaysia as a case study, this study develops a concept of HFSP. Since the literature on halal port is particularly scarce, it is important to conduct a preliminary study to identify the need of such framework. Therefore, the data presented in this paper were obtained from a comprehensive exploratory study conducted in the form of focus group discussion. The discussion was conducted with the ports' stakeholders, namely the shipping liners, forwarding agents, private jetty, as well as terminal and depot operators from the southern region of Malaysia. A total of 38 stakeholder companies (refer to Table 1) participated in the focus group discussion; they were divided into five groups led by a moderator in each group. The moderators were given a list of questions as a guideline, and the focus group discussion lasted three hours. The objective of the focus group discussion is to gain views on the feasibility of halal-friendly ports among the port stakeholders and, thus, determine the halal critical control points in the port operations.

**Table 1.** List of respondents from the focus group study.

| No | Category of Respondents | Position | Number |
|----|-------------------------|----------|--------|
| 1 | Shipping liners SAM | Branch Manager | 4 |
| | | Cargo Manager | 1 |
| | | Line Manager | 1 |
| 2 | Warehouse | Team leader | 2 |
| | | Warehouse Operations Head/Manager | 3 |
| 3 | Manufacturing companies | Senior Executive | 1 |
| | | Operation Executives | 5 |
| | | Assistant Supervisor | 1 |
| | | Product Engineer | 1 |
| | | Plant Administrator | 1 |
| 4 | Port operator | Senior Manager Business Development | 1 |
| | | Senior Manager Operations | 2 |
| 5 | Halal promoting agency | Senior Manager | 1 |
| 6 | Halal Park operator | Chief Executive Officer (CEO) | 1 |
| 7 | Private jetty operator | Operation Executive | 1 |
| | | Head of Operation | 1 |
| 8 | Logistics companies | Manager | 1 |
| | | Assistant Manager | 3 |
| | | Executive | 1 |

**Table 1.** *Cont.*

| No | Category of Respondents | Position | Number |
|----|------------------------|----------|--------|
| 9 | Container Depot Operator | Manager | 1 |
| | | Assistant Manager | 2 |
| | | Supervisor | 2 |
| 10 | Ports Shipping and Forwarding Association | Representative | 1 |
| | | TOTAL | 38 |

Prior to the focus group discussion, five initial interviews were conducted with the officers from operation, corporate, and marine units of the port authority to determine the initial halal critical control components or activities in the port operations. Thus, the information gained from these interviews formed the basis of the focus group discussion (refer to Figure 2 on the Research Methodology Process). As a result, four critical components of a successful HFSP were determined.

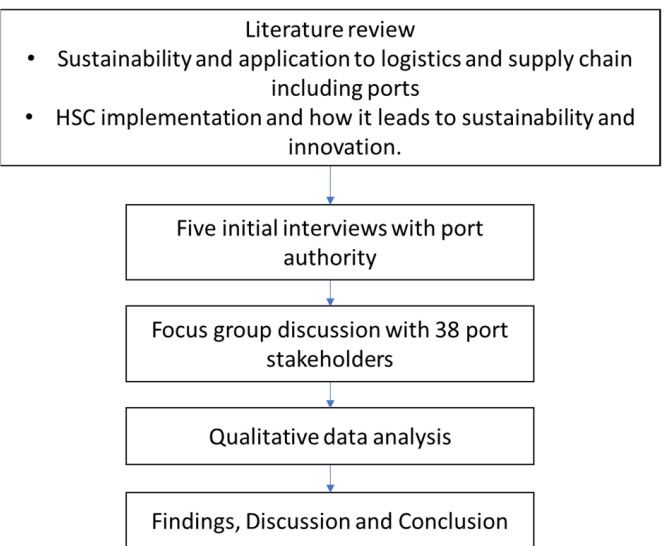

**Figure 2.** The Flow of Research Methodology Case Study.

Johor Port Authority (JPA), which is located at the southern tip of Malaysia, was employed as a case study for this research. JPA was established in 1975 as a statutory body established under the Port Authorities Act 1963. JPA launched its operations in 1977 and was the first port in Malaysia with a free trade zone. Since its formation, JPA has experienced a tremendous growth in 1995 leading to a private port operation under the Privatization Act 1990. Since then, JPA started to function as a regulatory body for the operations of the port operator, Johor Port Berhad (JPB). The port was located at the heart of the development of the 8000-acre Pasir Gudang Industrial Estate as well as spearheaded the new development of the 5000-acre Tanjung Langsat Industrial Complex (TLIC) and the Pengerang Integrated Petroleum Complex (PIPC). As a multipurpose port and the world's largest palm oil terminal, the port contributes as a fundamental driver of the Iskandar Malaysia's Comprehensive Development II, 2014–2025, under the Big Move 4 initiative and as the gateway to intra-Asia connectivity. Due to the need of meeting a high demand over a limited coastal area, the government has opened another private port company at the western region of the state, Port of Tanjung Pelepas (PTP), which began operation in October 1999. Both ports are strategically located at the Straits of Malacca, which is one of the world's busiest shipping lanes, connecting major economies, such as Japan, China, South Korea, and the Middle East. Thus, they positioned themselves as the Malaysia's

Southern Gateway. In fact, PTP is among the top list of world container ports, handling 9.8 million twenty-foot equivalent units (TEUs) in 2020.

As of today, Johor Port comprises of a total land area of 1000 acres and 660 acres in a Free Zone. It caters to 24 berths with a total berthing length of 4.9 km, which was designed for a capacity of 40 million tons. Currently, the ports provide five main services, namely container terminal, bulk and break-bulk terminal, liquid bulk terminal, warehouse facilities, and Pasir Gudang Port Free Zone. The major tenants include Lotte Chemical, Evergreen, Shell, Petronas, BASF, Allrig, FJB, Sibleco, Holcim, and Wilmar. The Halal Industrial Park in Johor, which is situated 13 km from the Johor Port, enhanced the Pasir Gudang area to be positioned as a potential halal hub as it is well connected to East Malaysia, China, Taiwan, Japan, Korea, South Asia, the Middle East, and South Africa.

## 4. Findings and Discussions

### 4.1. Development of Halal-Friendly Sustainable Port (HFSP) Model

Based on the initial interviews and focus group discussion, the respondents generally supported the halal-friendly port as a potential innovation in achieving sustainable practices. Being a Muslim majority country, most respondents are concerned of their responsibility to consume halal goods. Any halal issues should be addressed to ensure that Muslim consume true halal goods. They are also aware of the increasing demand of halal goods throughout the world and having a supply chain of halal goods is particularly crucial to fulfill the consumers' needs to obtain completely halal goods that comply to the Syariah principle throughout the entire supply chain process. However, they are not aware of the role that they may play in part of the halal supply chain implementation. In most countries, the halal implementation and certification are widely available for the manufacturing of food and beverages only. They do not realize that product contamination may also occur during the handling, transportation, and storing activities along the supply chain. In the context of international trade of halal products, the port handles several activities that may expose the halal products to contamination. Since the purpose of the halal practice is to avoid contamination of haram and hazardous elements, ports play a crucial role as an international gateway in the halal supply chain; thus, they need to comply to the halal requirements. Accordingly, they are eager to learn the roles they can play to fulfil these needs. Most participants in the focus group agreed that the implementation of halal logistics in the port area may address the contamination of halal cargoes handled in ports. The consensus on the feasibility of HFSP implementation among the focus group members led to the discussion on the identification of the criteria needed to establish HFSP. The reviews on the literature and interviews with the port authority prior to the focus group indicates four variables that contribute to the successful implementation of HFSP. The findings from the focus group supported that (1) dedication and segregation practices, (2) sanitation practices in the port operations, (3) determination of halal control points, and (4) traceability of the halal cargo (refer to Figure 3) are four that are vital components for HFSP establishment.

The components were developed based on halal practices that were derived from the Islamic guidelines that determine the sources of contamination. These can come from haram and harmful sources; thus, halal products can become haram (forbidden) and hazardous, and unsafe for consumption [91]. The lack of temperature control and the sanitation practices may also lead to the contamination of the cargo [94]. This scenario has led to the consensus among the participants of the following components determined as critical in the development of the HFSP model.

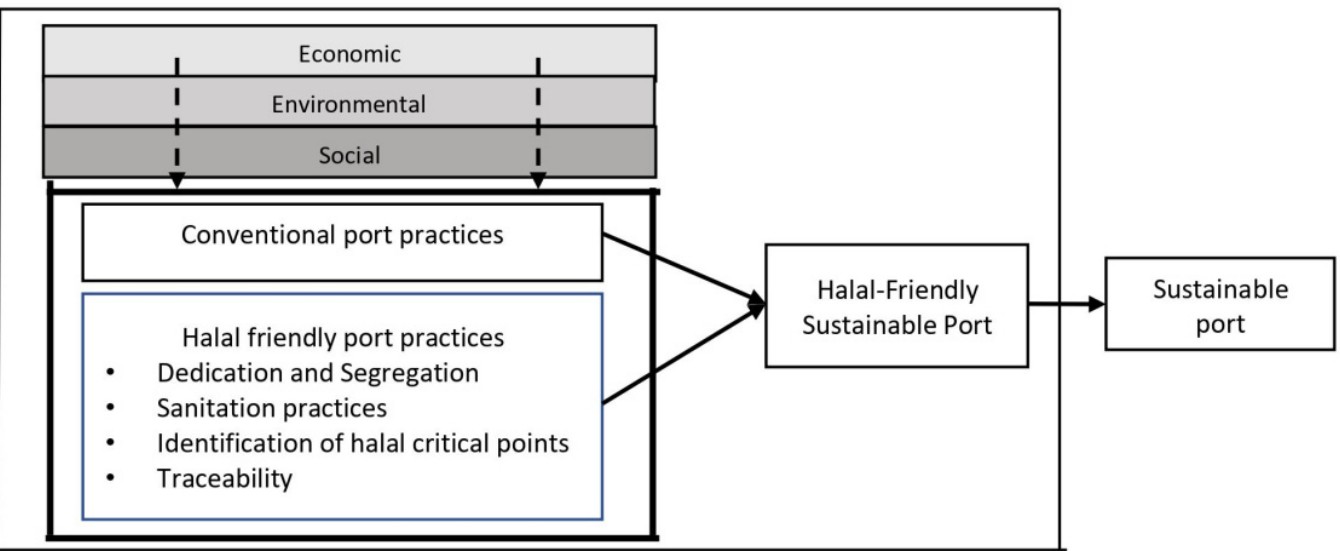

**Figure 3.** Halal-Friendly Sustainable Port Model.

### 4.1.1. Dedication and Segregation Practices

The findings demonstrate that the practice of HFSP should be based on two types of principle practices: segregation and dedication. When the logistics operations are solely dedicated for halal cargoes, there are less possibilities for the cargoes to be contaminated, thus reducing the number of halal control points. Halal control point is a point, step, or procedure, whereby control can be imposed, and contamination can be prevented and eliminated [115–117,119–121]. However, when the halal and non-halal cargoes or hazardous and non-hazardous cargoes are consolidated, the principle of segregation should be applied. This is because the possibilities of the cargoes to be contaminated are higher when these cargoes are consolidated. However, in handling the cold chain products, temperature should be controlled to avoid contamination in both situations. To implement a dedicated service, posts may allocate designated areas for halal declared cargoes. This dedicated service is feasible with full container load (FCL) types of shipment in comparison to less container load (LCL) because the container is loaded by halal certified shippers. However, in cases where it is almost impossible to dedicate halal cargo services, the principle of segregation should be applied.

These scenarios have become a major concern among the port stakeholders, especially regarding cost, which has led to several misconceptions on the feasibility of its implementation. They perceive that HFSP requires total transformation of the entire port process flow and may involve heavy investments when the rate of return of the investments is still uncertain. Although the paper is developed based on an ongoing study, the initial findings showed that the implementation of HFSP may involve minor adjustments of the port system, such as the declaration of halal goods in the documentation process, as well as physical cargo flow. HFSP is developed as a value-added service offered to the customer; therefore, the transformation of the entire port system is unnecessary. The additional line of halal cargo handling would not affect other port operations. The halal cargo handling may adopt similar practices to the handling of fragile and temperature-sensitive cargoes.

### 4.1.2. Sanitation Practices in Halal Logistics Operation

The standards of halal supply chain underline that cleanliness forms one of its vital components [115–117]. Thus, sanitation practices should be listed among the halal supply chain activities. Sanitation has been described by the World Health Organization as the provision of facilities and services for the safe removal of human urine and feces. It explains how hygienic conditions should be preserved through services, such as garbage collection and wastewater disposal. Marriot et al. [122] defined sanitation as "the creation and main-

tenance of hygienic and healthful conditions." Sanitation in halal logistics requirements covers the general requirements for premises, infrastructure, facilities, and workers.

The purposes of sanitation are as follows:

1.  To reduce contamination and promote the well-being of equipment, workers, and customers;
2.  To provide a healthy and pleasant environment;
3.  To protect the natural resources (e.g., surface water, groundwater, and soil) as well as to provide safety, security, and dignity for people when they defecate or urinate;
4.  To provide barriers between excreta and humans so that the chain of disease and infections could be stopped;
5.  To have and promote sustainable sanitation, particularly in developing countries;
6.  To meet regulatory environment.

In the context of this study, the suggestion on sanitation practices in port areas were well received by the participants, particularly at the warehouse areas.

### 4.1.3. Halal Control Point (HCP)

Following the above, to successfully implement the HFSP operations, the port operator needs to identify the halal control points that may cause contamination to occur on the halal cargoes. Throughout the process flow in the port area, this study found four halal control points, namely:

- Declaration of halal cargoes at the port entry point;
- Loading and unloading of cargoes;
- Storage and warehousing;
- Cargo inspection.

These halal control points (HCPs) are crucial as it associates with a certain degree of contamination risk towards the halal cargoes. The contamination refers to the contamination with haram and contamination with hazardous, or the halal cargo itself could be contaminated and become haram and hazardous due to lack of monitoring and control during the handling process. HCPs were identified based on guidelines underlined by the Islamic principles. The determination of HCPs along the process flow is vital to ensure that the risks associated with halal cargoes could be eliminated. To preserve the HCP, control measures need to be determined so that prevention may be conducted to reduce the risk of contamination. In any case of contamination occurrence, corrective actions that have been recognized earlier should be taken.

The consensus of non-conformity should be derived from the port operation staff and the Syariah (Islamic law) experts to ensure that halal practices are in line with the halal guidelines. The failure to comply with the standard operating procedure may lead to non-conformity of the process activities, such as the failure to declare the halal cargoes and thus subject to the conduct of corrective actions whenever possible.

### 4.1.4. Traceability

The establishment of an effective traceability system was found as crucial to ensure the quality and safety of the halal cargoes, and that product recall may be performed in any event of contamination. This traceability system would then act as an effective tool when the supply chain transparency is improved [123–125]. Several authors highlighted the lack of food traceability when the food products and production process information were often mishandled and missing in the companies, as well as among companies along the supply chain [126–129]. They urged for more detailed studies to be conducted at every stage of the supply chain process so that these processes could be documented, and the issues could be addressed [130–133]. None of the focus group members were against the fact that traceability drives an efficient halal supply chain within port operations.

Monitoring the movement of halal cargoes and its quality, as well as the process constraints throughout its flow in the port area, complements the safety and quality of the halal

cargoes in the entire supply chain. A system that can serve both internal and external supply chain traceabilities is crucial in the development of a fully traceable supply chain [125].

However, despite being regarded as a feasible operation, the requirements to be HFSP were viewed as a complex transformation process by some of the port stakeholders. Thus, they were not convinced with the newly introduced concept of HFSP due to the lack of evidence of a current successful halal port practice. Thus, the implementation of HFSP received low priority among port operators as they have chosen to focus on other strategic priorities of investment, such as digitalization and green practice towards meeting sustainable development goals (SDGs). However, most port stakeholders considered it as a new business model strategy to attract more port stakeholders to choose HFSP as a port of call due to new service offerings that address the SDGs.

### 4.2. Mapping the Implementation of HFSP to the Sustainable Development Goals (SDGs)

This study embedded all three components of the triple bottom line (i.e., environmental, economic, and social) to produce a sustainability value to the port. Table 2 maps the targets and indicators of the Sustainable Development Goals (SDG) that could be met by halal-friendly sustainable port.

**Table 2.** The halal-friendly sustainable port model's attainment of Sustainable Development Goals.

| Components | Goals | Targets | HFSP: Meeting the Targets of SDG |
|---|---|---|---|
| Economic | 3. Good Health and Well Being Ensures healthy lives and promote well-being for all at all ages | 3.9 By 2030, substantially reduce the number of deaths and illnesses from hazardous chemicals and air, water and soil pollution and contamination. | The foundation of halal practice (i.e., to avoid contamination) may ensure that final goods that reach consumers are safe to be consumed and that the quality may promote health and well-being of the society. |
| Economic | 9. Industry, Innovation and Infrastructure Build resilient infrastructure, promote inclusive and sustainable industrialization and foster innovation. | 9.1 Develop quality, reliable, sustainable and resilient infrastructure, including regional and transborder infrastructure, to support economic development and human well-being, with a focus on affordable and equitable access for all. | As one of the crucial transport infrastructures, ports could promote industry innovation through the development of value-added services of halal handling that could offer choices to the halal certified manufacturers. |
| Social | 10. Reduced Inequalities Reduce inequality within and among countries | 10.2 By 2030, empower and promote the social, economic and political inclusion of all, irrespective of age, sex, disability, race, ethnicity, origin, religion, or economic or other status. | Addressing the needs of the Muslims dietary requirements that demand them to consume halal food may encourage inclusion of all irrespective of religion. |
| Environment | 12. Responsible Consumption and Production Ensure sustainable consumption and production patterns | 12.3 By 2030, halve per capita global food waste at the retail and consumer levels and reduce food losses along production and supply chains, including post-harvest losses. 12.8 By 2030, ensure that people everywhere have the relevant information and awareness for sustainable development and lifestyles in harmony with nature. | The reduction in contamination may reduce waste, thus indicating the ability of the model to complement that part of supply chain to reduce food waste and food losses along production and supply chains. Halal-friendly sustainable port forms part of the halal supply chain may create awareness among the society on the importance to have a sustainable lifestyle that move in tandem with nature when the supply chain of halal is completed from end to end. |

| Components | Goals | Targets | HFSP: Meeting the Targets of SDG |
|---|---|---|---|
| Social | 17. Partnerships for the Goals: A successful sustainable development agenda requires partnerships between governments, the private sector and civil society. These inclusive partnerships built upon principles and values, a shared vision, and shared goals that place people and the planet at the centre, are needed at the global, regional, national and local level. | 17.11 Significantly increase the exports of developing countries, with a view to doubling the least developed countries' share of global exports by 2020. | Implementing Halal-friendly sustainable port requires cooperation and collaboration among various stakeholders in the port community. Thus, it supports the country aspirations to be the Global Halal Hub through sufficient capacity handling for halal domestic and export cargo. |

From an economic perspective, the aim of avoiding contamination from haram and hazardous cargo in the port area may assist in achieving target 3.9 under Goal 3, i.e., to ensure healthy lives and promote well-being for all ages. Ports, as one of the crucial transport infrastructures at the border, also play a key role in economic perspectives to achieve target 9.1 from Goal 9. The new value-added services offered to the customers in addition to the current port services (refer to Figure 2) may enhance the development of quality, reliability, sustainability, and resilience of the port to support the economic development. As more halal certified companies emerge in the market, the demand of halal cargo handling will also materialize. HFSP will provide new service offerings that provide choices to their customer, who would like to have a specific service of halal cargo handling, transportation, and storage, leading to service fulfilment.

The reduction in contamination rates may reduce waste, indicating the ability of the model to complement that part of the supply chain to meet the environment target 12.3, particularly by reducing food waste and food losses along the production and supply chains. By introducing HFSP as part of the halal supply chain, the society will grow more and more aware of the importance of having a sustainable lifestyle that moves in tandem with nature when the supply chain of halal is completed from end to end and can lead to the achievement of target 12.8.

Accordingly, the social element in this model is reflected in the responsibility of the port in ensuring that the halal cargo handling is free from contamination with haram and hazardous substances, hence influencing the customers' trust in the port operator. The halal certification holds the branding that guarantees the quality of port services as it reflects its compliance to the halal supply chain standard guideline. Accordingly, it aims to meet target 10.2, i.e., empower and promote the social, economic, and political inclusion of all, irrespective of religion or other status. Thus, the model addresses the crucial requirement of the increased number of Muslims worldwide in accommodating their dietary needs that may also benefits others due to the halal product quality. Implementing a HFSP requires cooperation and collaboration among various stakeholders in the port community. Thus, it supports the country's aspirations to be the Global Halal Hub through sufficient capacity handling for domestic and export cargo, leading to the attainment of target 17.11.

The assimilation of all these sustainability elements qualifies the model to be called a HFSP.

## 5. Research Implications

This study contributes to the body of knowledge through the development of a HFSP model. Specifically, it extends the current halal supply chain literature that comprises of halal transportation, halal warehousing, and halal retailing. The current components of halal supply chain seemed applicable only to the domestic context of a country, thus pushing the need to include the international gateway context. Globalization has led

the production and supply of halal products being located across the globe so as its consumption. Various issues occur throughout the journey because a lengthier journey may be associated with more complex procedures and exposure to contamination. Accordingly, ports have the chances to work closely with their clients in creating innovation projects that may fulfill their customer desires [19,134]. Based on this call, this study proposes a HFSP model to complement the current practice of halal supply chain.

The model specifies the presence of sustainable components (environmental, economic, and social), indicating that halal practices are sustainable practices. The lack of research in this area implies the new knowledge discovered leading to innovation. Calic et al. [114] stressed that innovation can be in the form of providing the focal organization with goods and services that add new forms of value to clients. Flint et al. [19] emphasized that any logistics service that addresses both internal and external operations that are viewed as new and useful to the customers is considered as logistics innovation. This innovation enhances operational efficiency and fulfils customer requirements. Thus, the concept of HFSP can be considered as an innovation because the service of handling of halal cargoes in the port area is a newly invented service that offers the certified halal customers an extra value in preserving the quality and safety of the halal cargoes throughout the journey. Schaltegger et al. [12] proposed that sustainability plays an essential role in business organizations in driving excellent performance through the innovation development.

In ensuring a successful halal-friendly sustainable port implementation, this study found four main variables that are considered crucial: (1) dedication and segregation practices; (2) sanitation practices in the port operations; (3) determination of halal control points; and (4) traceability of the halal cargo (refer to Figure 3). However, more qualitative data will be collected to refine the model. Thus, other variables and the detailed process flow of the port operations need to be explored and identified. The variables might also be different depending on the types of port operation that might impose varying degrees of contamination risk.

From an industry perspective, the significant growth of global halal market has attracted many logistics service providers and ports to embrace halal supply chain spirit and practices so that they can stay competitive. With the major Muslim market in Malaysia and the world's largest Muslim population neighbor market in Indonesia, there is huge potential within market opportunity. The remarkable development of the halal supply chain standards offered by Standards Malaysia provides a strong support for the halal supply chain implementation. Following the adoption of these standards at the Standards and Metrology Institute for the Islamic Countries (SMIIC), Turkey has opened more opportunities for the halal supply chain certification adoption at an international level. The fact that ports are considered as a network of business success may lead to the whole supply chain competitiveness [135–138] and reflect the crucial requirements of a joint effort between port authorities and logistics firms in addressing the sustainability and competitive challenges.

## 6. Limitations and Direction for Future Research

The results presented in this study were based on the comprehensive exploratory stage of an ongoing study in the development of HFSP model. As a continuation, the model will be refined after more qualitative data are collected. In-depth interviews will be conducted with the Islamic law experts, port industry practitioners, and academic experts of the feasibility of the model as well as its implementation. Current scenarios have shown the occurrence of various halal issues at the international gateway, particularly ports in Malaysia. This is because, generally, practicing Muslims are very sensitive to the halal-related issues, especially products that they consume, such as food and pharmaceuticals. The only reason is that the consumption of halal food is an order from Allah SWT and those who do not obey will be considered as committing a sin.

Having considered the above, further studies are encouraged to explore the costing aspect of the halal operations as it involves a value-added service offered to the customers. Specifically, the costs and benefits may be calculated in quantitative terms to ensure the

feasibility of the implementation. This is because the details of how a halal port operates are necessary to ensure that they propose the precise value to the customers. Fulfilling the responsibility to preserve the quality and safety of products totally depends on the integrity of each party along the supply chain. Thus, this would also provide avenues for further research.

**Author Contributions:** Conceptualization, H.S.J., M.R.A., M.L.A.A. and N.F.; Methodology, M.L.A.A., M.R.A. and H.S.J.; Investigation, M.L.A.A. and H.S.J.; Formal analysis, H.S.J. and N.F.; Resources, M.L.A.A. and M.R.A.; Writing—original draft preparation, H.S.J. and M.L.A.A.; Writing-review and editing, H.S.J., M.R.A., M.L.A.A. and N.F.; Project administration, M.L.A.A. All authors have read and agreed to the published version of the manuscript.

**Funding:** This research no external funding.

**Institutional Review Board Statement:** Not applicable.

**Informed Consent Statement:** Informed consent was obtained from all subjects involved in the study.

**Data Availability Statement:** Not applicable.

**Acknowledgments:** The authors would like to thank Johor Port Authority for their support and cooperation on the research.

**Conflicts of Interest:** The authors declare no conflict of interest.

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
