# Peer review of "Creating Innovation in Achieving Sustainability: Halal-Friendly Sustainable Port"

_sustainability, doi:10.3390/su132313339_

Round 1

Reviewer 1 Report

The paper presents an interesting topic regarding special port operations of "halal" logistics. I am aware of the increased value of halal products and processes which are basically quality management and ensure.

The authors highlighted that they applied respondents from various stakeholder groups (Table 1) also stressed that their consensus is a precondition of sustainable long-term operations. In the discussion, however, we do not see how the responses differed among the stakeholders. Did they provide conflicting responses in some cases? Were there extreme opinions in any of the representatives? Creating a consensual result in the case of diverse stakeholder opinions is a significant topic in management sciences.

My other concern is that implementing halal logistics operations causes less interoperability in the services. Logistics is cheaper when the haulage, the loaded products, and containers are standardized and handled together without making any distinctions. What will be the cost impact of halal types of transport and operations? This should be discussed in detail in the revised paper.

Minor comments:

Please improve Figure 2, some of the text cannot be seen.

Some English corrections are necessary, e.g. much lesser.

Author Response

Dear Reviewer,

Reviewer 2 Report

The topic is good and novel. I can see that you had done lots of work to analyze sustainability on Halal- Friendly Sustainable Port. Meanwhile, there are some problems to be solved in it. Below are some suggestions:

Firstly, key words of the paper were too much for 12, maybe, it would better for 5; yours research should focused on the Key content, please thinking about the objectives in paper.

Secondly, the second part (Background of the Study) of the paper was too much(L65-L348). but the third part(L349-L491) was just too little, which should be more to discussed. it just seems that describes the Islamic guidelines. what effective in the application of port? Are there any improvements in operation? the third part was inadequate discussion.

Thirdly, fig2 is not clearly. The language should be improved.

Also, there are still some other details worth considering in this paper.

I hope the suggestions helpful to you.

Author Response

Dear Reviewer, 

Please find attached our reply to your comments.

Many thanks.

Reviewer 3 Report

Some suggestions that can improve the quality of the paper:

  1. There is a lack of description of paper structure at the end of Introduction.
  2. To improve the quality of the paper I would like to propose the Authors to add a graphic scheme as a presentation of the proposed methodology.
  3. The value of the article would be much greater if the authors introduced an extension of discussion elements and comparisons with the results of other studies.
  4. Moreover, the application of the research  should be more detail explained. The results and  discussion of the results are very general. 
  5. Finally, please improve the quality of figure 2 (not readable).

Author Response

Dear Reviewer,

Please find attached our reply to yopur comments.

Many thanks.

Round 2

Reviewer 1 Report

The authors have successfully completed the revision, all of my main concerns have been covered. Thus, I recommend the paper for publication.

Reviewer 2 Report

none

Reviewer 3 Report

I have carefully checked the revised version of the manuscript and also and authors responses to reviewer comments. The revised version is improved and all reviewer comments were taken under consideration, so I recommend the paper for publication.